# Polyphenol-Rich *Aronia melanocarpa* Fruit Beneficially Impact Cholesterol, Glucose, and Serum and Gut Metabolites: A Randomized Clinical Trial

**DOI:** 10.3390/foods13172768

**Published:** 2024-08-30

**Authors:** Morgan L. Chamberlin, Jesse T. Peach, Stephanie M.G. Wilson, Zachary T. Miller, Brian Bothner, Seth T. Walk, Carl J. Yeoman, Mary P. Miles

**Affiliations:** 1Department of Food Systems, Nutrition, and Kinesiology, Montana State University, Bozeman, MT 59717, USA; morganchamberlin@montana.edu (M.L.C.);; 2Department of Chemistry and Biochemistry, Montana State University, Bozeman, MT 59717, USA; 3Department of Food Chemistry and Toxicology, Faculty of Chemistry, University of Vienna, 1090 Vienna, Austria; 4United States Department of Agriculture, Agricultural Research Service Western Human Nutrition Research Center, Davis, CA 95616, USA; 5Institute for Advancing Health through Agriculture, Texas A&M, College Station, TX 77845, USA; 6Department of Research Centers, Montana State University, Bozeman, MT 59717, USA; 7Department of Microbiology and Cell Biology, Montana State University, Bozeman, MT 59717, USA; 8Department of Animal and Range Sciences, Montana State University, Bozeman, MT 59717, USA

**Keywords:** Aronia, gut microbiome, dietary bioactives, metabolomics, polyphenols, inflammation, functional food

## Abstract

Polyphenol-rich Aronia fruits have great potential as a functional food with anti-inflammatory, hypolipidemic, and hypoglycemic biologic activities. However, clinical intervention trials investigating the impact of Aronia fruit consumption on human health are limited. A randomized, controlled, double-blinded, parallel intervention trial was conducted using 14 human subjects who ingested either 0 mL or 100 mL of Aronia juice daily for 30 days. Anthropometric measurements, fasting, and postprandial measures of glucose and lipid metabolism and inflammation, 16S rRNA fecal microbial composition data, and mass spectrometry-acquired serum and fecal metabolomic data were collected before and after the intervention period. Data were analyzed using general linear models, ANOVA, and *t*-tests. Daily consumption of Aronia prevented a rise in cholesterol levels (β = −0.50, *p* = 0.03) and reduced postprandial glucose (β = −3.03, *p* < 0.01). No difference in microbial community composition by condition was identified at any taxonomic level, but a decrease (β = −18.2, *p* = 0.04) in microbial richness with Aronia was detected. Serum and fecal metabolomic profiles indicated shifts associated with central carbon and lipid metabolism and decreases in pro-inflammatory metabolites. Our study further informs the development of polyphenol-based dietary strategies to lower metabolic disease risk.

## 1. Introduction

Non-communicable, chronic diseases including cardiovascular disease and type 2 diabetes are among the highest contributors to global morbidity and mortality rates [1]. Low-grade inflammation [2], cellular damage from oxidative stress [3,4], insulin resistance [5], and impairment of carbohydrate and lipid metabolism [6] contribute to the pathogenesis of these diseases. Polyphenols are dietary bioactive compounds found in a wide variety of foods that exhibit anti-inflammatory and antioxidant activity by preventing activation of pro-inflammatory cellular mediators and reactive oxygen and nitrogen species [7,8,9,10] and regulating antioxidant enzyme concentrations [7]. Polyphenols can also improve systemic markers such as blood pressure and lipid profiles [11], which has prompted interest in polyphenol-rich food consumption as a dietary strategy to combat chronic disease progression.

*Aronia melanocarpa*, or chokeberry, is a hardy, cold weather shrub which produces fruit rich in dietary polyphenols. Aronia fruits are the richest fruit source of polyphenols, primarily derived from anthocyanins, procyanidins, and hydroxycinnamic acids, with 1756 mg/100 g as determined by chromatography methods [12]. The high polyphenol content of Aronia fruit confers high antioxidative potential—ranked the second highest of fruits after elderberries with 1752 mg/100 g [12]. The health promoting potential of polyphenol-rich Aronia fruits defines them as a functional food. This recognition has led to the widespread commercial production of Aronia-based products including supplements, juices, preserves, and teas. However, there are limited human studies examining the metabolic benefits of Aronia fruit consumption, and the results are conflicting, with some studies citing improvement to measures of inflammation and lipid and glucose metabolism and other studies reporting no effect [13,14,15,16,17,18]. These conflicting results may be due to dose, duration, or delivery method. Differences in metabolic health status may have also contributed to large variability in study results.

Variability in host gut microbial communities may also contribute to conflicting study results. Polyphenol bioavailability is influenced, in part, by the physicochemical properties of the compound while polyphenol bioefficacy is likely mediated by the gut microbiome [19,20,21,22]. Although some polyphenols are absorbed in the small intestine, most pass to the large intestine for microbial enzymatic action. Gut bacteria can metabolize polyphenols into phenolic metabolites [23,24]. The bioavailability and bioefficacy of microbial-produced phenolic derivatives frequently exceed the parent compound [24,25,26,27]. Furthermore, evidence suggests that polyphenol intake can alter microbiome composition in an anti-microbial and prebiotic-like manner [28], which has the potential to influence the gut metabolome [22] and host post-prandial responses [29]. Therefore, changes in microbial abundance and gut-derived metabolites represent additional mechanisms of action to achieving polyphenol-associated health benefits [19]. However, there is a paucity of research investigating the influence of Aronia consumption on the gut microbiome in adults.

Furthermore, while the importance of postprandial metabolism in relation to metabolic health and disease risk is well established [30,31,32,33,34], these measures are not consistently analyzed in non-acute polyphenol intervention studies. Clinical trials investigating how polyphenol-rich foods influence postprandial metabolism and the fecal microbial community are needed to accurately assess their efficacy to beneficially impact health. We therefore completed a 30-day clinical intervention trial in healthy adults to assess the impact of Aronia fruit juice consumption on host metabolism, fecal microbial composition, and host and microbial metabolomic profiles. Our research strategy was to perform a feasibility study to comprehensively analyze the impact of Aronia in a small cohort of healthy individuals to identify robust impacts for further investigation. In this exploratory investigation, we assessed both fasting and postprandial lipidemic, glycemic, and inflammatory responses to long-term Aronia consumption as well as changes to microbial taxonomy through 16S rRNA sequencing and serum and gut metabolites. We hypothesized that Aronia consumption would improve markers of systemic metabolism and shift gut microbial composition and/or microbial metabolism.

## 2. Materials and Methods

### 2.1. Ethics Statement

The protocol was approved by the Montana State University Institutional Review Board (MC010819). Verbal review of the informed consent was completed in person, and time was allotted for questions and clarifications. All participants completed written informed consent prior to their participation. The study was retrospectively registered February 2022 at ClinicalTrials.gov (NCT05255718).

### 2.2. Study Population

Potential participants were recruited on a rolling basis in Bozeman, Montana from April 2019 through September 2019 via flyer advertisements, word of mouth, email lists, and snowball recruitment methods. Interested individuals were screened for study eligibility by email or phone. Potential participants were 18–55 years of age and were excluded for chronic disease diagnosis, regular use of anti-inflammatory medications, use of blood pressure or lipid-lowering medications, use of antibiotics in the prior 90 days, allergies to wheat, chokeberry, and/or dairy, or being pregnant. Participants were not excluded for any other dietary, lifestyle, or health history criteria.

### 2.3. Research Design

The study was a double-blind, randomized, placebo-controlled intervention trial designed to assess the impact of chronic Aronia juice consumption. Participants consumed either 100 milliliters (mL) of Aronia juice (ARO) or a placebo juice (PLA) per day for 30 days. The placebo was low in polyphenol content and matched in taste and color using food colorants and artificial flavoring. Testing took place at the Montana State University Nutrition Research Laboratory. Participants completed three data collection visits at the lab. Subject consent and anthropometric measurements were obtained at visit 1. Visit 2 occurred after a one-week diet washout period during which participants were asked to omit foods with high polyphenol content from their diet. A high-fat meal challenge (buttered toast) with postprandial blood collection for glucose, lipid, inflammation, and metabolomic analyses was performed at visits 2 and 3 with a 30-day intervention period in between. Participants brought a self-collected stool sample to these visits for fecal microbial taxonomy and metabolomic analyses. A two-day timeframe was incorporated into the intervention period, allowing for a potential intervention duration ranging from 28 to 32 days. Participants consumed their last juice dose the day prior to their final visit. Participants were instructed to omit foods with high polyphenolic content from their diet for the duration of the study.

### 2.4. Anthropometrics

Participant anthropometric measurements were collected at baseline and again after completing the intervention. Subjects were instructed to refrain from eating, drinking, or exercising in the 3 h prior to testing. Participant height was measured using a stadiometer. A validated [35] segmental multifrequency bioelectrical impedance analysis (SECA mBCA 515, Hamburg, Germany) was utilized to measure weight (kg), fat mass (%), and visceral adipose (L). Blood pressure was taken in the mornings of visits 2 and 3 prior to the high-fat meal challenge after subjects had been seated for at least 15 minutes (min). Two automated measurements were taken with the mean used for analysis.

### 2.5. Randomization

Block randomization was performed using the blockrand function in the R blockrand package [36]. Three blocks with two levels were created, and a seed for each block was created using the last five integers generated by the sys.time function. Participants were randomized to the Aronia (ARO) or placebo (PLA) group after completion of the first visit. Randomization and juice preparation was performed by a designated research team member who did not interact with participants. Investigators performing research visits were not aware of the treatment allocation. After study completion, the juice condition was unblinded for data analysis.

### 2.6. Intervention

Participants were asked to drink 100 mL daily of ARO or PLA juice during the 30-day intervention period. This intervention dose was based on prior studies utilizing a similar or smaller anthocyanin dose demonstrating decreased postprandial lipid and inflammation response to acute intake [37] and decreased inflammation markers with chronic intake (~30 days) in healthy humans [38]. Timing of juice intake was not enforced but participants were encouraged to consume juice at roughly the same time every day to improve compliance. Raw juice was heat pasteurized before study use. The PLA juice was matched for color, flavor, and carbohydrate content to ARO juice and consisted of 128.5 grams (g) of sorbitol, 74.5 g of glucose, 77.9 g of fructose, 28.8 g of black cherry Koolaid© mix (no sugar added), 4 ounces of lemon juice, 16 drops of blue food coloring, and enough water to create 1 liter (L) of solution. In another polyphenol intervention study, the use of black cherry Koolaid in a placebo drink was found to provide minimal polyphenol content and antioxidant activity [39]. Participants returned to the laboratory approximately three times during the intervention to receive a fresh 1 L supply of juice provided in a half-gallon plastic jug. Plastic jugs were placed into labeled paper bags and refrigerated until participant pickup.

During the intervention period, participants were instructed to consume their standard diet with the exception of omitting foods with high polyphenolic content including certain fruits and vegetables such as berries and kale, red and black beans, and dark chocolate. The full list of foods is provided in Appendix A. They were instructed to follow this diet for a one-week washout period prior to their second visit as well as the subsequent intervention period for a total of approximately five weeks. Adherence to diet guidelines was verified with a self-reported diet adherence questionnaire completed during the one-week washout period as well as the intervention period.

### 2.7. Aronia Juice LCMS and NMR Analysis

Aronia juice was provided by the Western Agricultural Research Center and consisted of a blend of Viking, Mackenzie, and Autumn Magic cultivars grown at the center in Corvallis, Montana. The Aronia juice was analyzed to determine both carbohydrate and phenolic composition, as previously described [22]. In brief, polyphenol content was determined using a targeted liquid chromatography–mass spectrometry (LCMS) method developed for use on an Agilent 6538 quadrupole time-of-flight (Q-TOF) mass spectrometer (Agilent, Santa Clara, CA, USA) and an Agilent 1290 ultrahigh-performance liquid chromatography (UPLC) system (Agilent, Santa Clara, CA, USA). An Acquity HSST-3 UPLC reverse phase column, 1.8 μM, 100 mm/2.1 mm (Waters, Milford, MA, USA), was utilized to achieve separation. The juice carbohydrate content was determined by nuclear magnetic resonance (NMR) analysis using a Bruker 600 MHz Avance III NMR spectrometer (Bruker, Billerica, MA, USA) with a 600 MHz TCI (H-C/N-D05Z) LT probe. Determined carbohydrate concentrations were used to develop matching placebo juice. Carbohydrate and phenolic concentrations of the Aronia juice are provided in Table 1.

### 2.8. Habitual Diet Assessment

Habitual diet was assessed using the validated diet history questionnaire (DHQ III), a web-based survey offered by the National Cancer Institute. While seated during the high-fat meal challenge at visit 2, participants were asked to complete a 30-day recall. Participants were instructed to provide best estimates of habitual intake with the acknowledgment that precise recall of portion sizes for all beverage and food consumption is unlikely. The DHQ III survey includes 26 supplement and 135 food and beverage questions to assess dietary frequency and portion sizes. Healthy Eating Index (HEI, 2015) scores were calculated through built-in analysis of the DHQ III surveys. These scores serve as an indication of diet quality in compliance with the 2015–2020 US Dietary Guidelines for Americans [40,41]. The analyzed HEI scores included nine scores based on adequacy (whole grains, fatty acids, total fruit, greens and beans, whole fruit, total vegetables, seafood and plant proteins, dairy, and total protein foods) and four scores based on moderation (added sugars, refined grains, saturated fats, and sodium). The total HEI score, representing the sum of all scores, was also analyzed.

### 2.9. High-Fat Meal Challenge

The high-fat meal was made by spreading 58.3 g of salted butter (Tillamook, Tillamook, OR, USA) on three pieces of toasted whole wheat bread (Wheat Montana, Three Forks, MT, USA). In total, this meal contained 714 kilocalories (kcal), 12 g of protein, 54 g of carbohydrate, 50 g of fat, and 9 g of dietary fiber. Ad libitum water was provided for all participants and caffeinated Earl Grey black tea (Bigelow, Fairfield, CT, USA) was provided for participants who identified as habitual coffee consumers. Timing of the postprandial period began when participants started the meal and they were given 15 min to finish eating.

### 2.10. Blood Sampling

Participants were instructed not to participate in strenuous physical activity or consume any alcohol in the 24 h prior to their second and third visits and to complete an overnight (10–12 h) fast. Both visits took place in the morning. Participants were instructed to consume their last juice dose the day prior to the third visit. A certified phlebotomist collected blood samples by antecubital venipuncture. A fasting (baseline) blood sample was collected before participants consumed the high-fat meal followed by additional blood collections at hours 1, 2, 4, and 6 post meal consumption. Blood was collected into EDTA, serum separating, and heparinized vacutainer tubes (BD Vacutainer, Franklin Lakes, NJ, USA). Serum separating tubes were allowed to clot for 15 min at room temperature before centrifugation (1200 RPM, 15 min). Serum aliquots were frozen at −80 °C until analysis. Heparinized and EDTA-additive whole blood was used directly after collection for blood marker analysis.

### 2.11. Analysis of Blood Markers

Low-density lipoprotein cholesterol (LDL), high-density lipoprotein cholesterol (HDL), total cholesterol (CHOL), triglyceride (TG), and glucose (GLU) levels were determined from heparinized whole blood using Picollo Xpress Chemistry Analyzer lipid panels (Abaxis, Union City, CA, USA). Whole blood collected in an EDTA tube was used to determine glycated hemoglobin (HbA1c) using an Affinion2 analyzer (Abbott, Princeton, NJ, USA).

### 2.12. Analysis of Inflammation Biomarkers

High-sensitivity multiplexing (Bio-Rad Bio-Plex^®^ 200 HTS, Hercules, CA, USA) was utilized to measure serum cytokine levels following Millipore (EMD Millipore Corporation, Billerica, MA, USA) procedures. Analyzed cytokines included granulocyte macrophage colony stimulating factor (GM-CSF), interferon-gamma (IFN-γ), tumor necrosis factor alpha (TNF-α), and five interleukins (IL) including IL-1β, IL-6, IL-10, IL-17, and IL-23. Testing was performed on two samples from each time point and the mean was used for analysis. Values measured as under the detectable limit were replaced with ½ the minimum detectable concentration for the corresponding cytokine for analysis [42].

### 2.13. Stool Collection

Participants were asked to collect two fecal samples in total for determination of the gut microbiome composition and gut metabolome. Self-collection kits and written instructions were provided to participants for self-collection of a stool sample within 24 h of their baseline and final visits. Stool samples were collected into a sterile 50 mL Eppendorf tube and transported under refrigeration to researchers. Upon arrival for their study visit, samples were aliquoted for 16S rRNA and fecal metabolomic analysis. Stool aliquots were processed with pre-reduced phosphate-buffered saline and aliquoted into cryogenic vials in an anaerobic chamber. Samples were stored at −80 °C until further processing for DNA and metabolite extraction.

### 2.14. Genomic DNA Extraction and Microbial Analysis

Dneasy Powersoil^®^ Pro Kits (QIAGEN Sciences, Germantown, MD, USA) were utilized for bacterial DNA extraction from human fecal samples. Extracted DNA was stored at −80 °C until analysis. Illumina MiSeq amplicon sequencing of the 16S rRNA V4 region was performed by the University of Michigan, Michigan Microbiome Project. After DNA quantification, V4 amplicon libraries were generated with dual-index barcoded primers, then by library purification, pooling and MiSeq paired-end (2 × 250) sequencing. MOTHUR software (Version 1.35.1) (mothur.org) was used to process and curate sequencing reads following standard MOTHUR operating procedures for the MiSeq platform [43]. In brief, contiguous sequences were assembled from paired-end reads and screened for length and quality. Remaining contigs were aligned to the SILVA ribosomal RNA database (Release 132) to improve alignment of rRNA sequences. The UCHIME algorithm in MOTHUR was utilized to identify and remove potentially chimeric sequences. The Bayesian classifier of the Ribosomal Database Project was used to assign taxonomic classifications. After non-target-read removal, operational taxonomic units (OTUs) were assigned using VSEARCH distance-based clustering with a 97% similarity threshold.

### 2.15. Serum Metabolite Extraction

Serum samples were stored at −80 °C until LCMS analysis, when samples were removed from storage and allowed to thaw on ice. A liquid extraction was completed by first removing 20 µL of serum, which was placed in a clean vial with 80 µL of cold acetone to precipitate serum protein. Precipitation was aided by storing vials at −80 °C for two hours. Samples were then spun in a centrifuge at 20,000× *g* for 30 min to pellet protein. The resulting metabolite supernatant was collected in a clean vial and concentrated in a vacuum concentrator until dry. Dry samples were stored at −80 °C until ready for LCMS analysis. Directly prior to LCMS analysis, metabolites were reconstituted with 40 µL of MeOH:H_2_O (50:50) and placed in a clean LC vial.

### 2.16. Fecal Metabolite Extraction

Fecal analysis was undertaken by first weighing out 1 g of fecal matter which was stored at −80 °C until ready for analysis. Fecal matter was thawed on ice and three volumes of cold water were added. The fecal slurry vial was next placed in cold zinc beads and sonicated for five min at a 40% duty cycle using a sonication probe. After sonication, an equal volume of methanol was added to the slurry and agitated for one minute. Samples were then stored at −80 °C for one hour. Samples were agitated again for one minute and spun in a centrifuge at 20,000× *g* for 15 min. The supernatant was removed and placed in a clean vial. An equal volume of MeOH:H_2_O (50:50) was added to the pellet. The same sequence of sonication, cold storage, agitation, and centrifugation was repeated as a washing step. The resulting supernatant layer was then removed and added to the first layer.

A protein precipitation was next completed using acetone. Five sample volumes of acetone were added to the sample tube, after which samples were stored at −80 °C for 12 h. After storage, samples were spun in a centrifuge at 20,000× *g* for 5 min to pellet protein. The metabolite-rich supernatant was removed and placed in a clean vial. The samples were then concentrated using a vacuum concentrator to dryness and stored at −80 °C until LCMS analysis, at which time they were reconstituted in 40 µL of MeOH:H_2_O (50:50).

### 2.17. LCMS Metabolomics Analysis of Fecal and Serum Extracts

LCMS analysis was completed at Montana State University in the Proteomics, Metabolomics and Mass Spectrometry Facility. Analysis of serum metabolites was completed using an Agilent 1290 UHPLC system coupled to an Agilent 6538 Q-TOF MS. All runs were completed in positive mode and separation was achieved using a 130 Å, 1.7 μm, 2.1 mm × 100 mm Acquity BEH-HILIC HPLC column with water as mobile phase A and acetonitrile as mobile phase B, both with 0.1% formic acid. The solvent gradient began at 90% A for one minute, and then, decreased linearly to 60% A at five minutes before returning to 90% A for an additional minute. The total analysis time was 6 min, with a constant solvent flow rate of 400 µL/min and a column temperature of 30 °C. Identifications were completed using authentic standards from an in-house library, fragmentation matching using MS-DIAL [44] and the MoNA spectral library, and in silico MSMS annotation using SIRIUS and CSI:Fingerprint software (version 5) [45]. MS-DIAL annotation parameters included a 10 parts per million (ppm) error window and an identification score cutoff of 50%. SIRIUS parameters included a 10 ppm error window and matches were accepted only if they had a COSMIC score over 0.4 and only the first structure was selected. MSMS analytical injections were completed using identical LCMS conditions.

Fecal samples were analyzed using a Waters Synapt-XS Q-IMS-TOF and a Waters I-Class UHPLC. Ionization was completed in negative mode using ESI with a 100 Å, 1.8 μm, 2.1 mm × 100 mm Waters Acquity HSST3 reverse phase column. The column compartment was kept at 40 °C and the flow rate was 200 µL/min. The analysis runtime was 28 min and used UHPLC-grade water and acetonitrile, each with 0.1% formic acid, as mobile phases A and B, respectively. Analysis began with 99% A until 0.5 min, at which time A began to decrease linearly to 1% at 25 min. At 25 min, A increased to 99% and was held there until the end of the run. Identifications were completed using authentic standards and isotopic scoring and fragmentation matching using MassLynx referencing the HMDB spectral library [46]. Features with combined scores greater than 30 were retained as positive annotations. MSMS analytical injections were completed using identical LCMS conditions.

### 2.18. Statistical Methods

Analysis was conducted in RStudio (1 June 2023) using R 4.3.1 and data were visualized using the ggplot2 [47] and effects [48] packages.

#### 2.18.1. Power Analysis

Power was calculated a posteriori using untargeted serum metabolomic data from fasting blood samples at pre- and post-intervention time points. The top metabolites (features) over time and the intervention group were identified with multivariate empirical Bayes analysis of variance (MEBA) [49]. MEBA is a time-series modeling technique that compares temporal metabolic profiles across different conditions and ranks features based on degree of change over time. The features identified by MEBA analysis were ranked from most significant to least based on differences in patterns over time. The features from this list were then binned into ten groups (10th percentiles) to capture a range of most to least effective features based on how well they discriminated between intervention groups over time. To quantify the effect of selected features, Cohen’s d effect sizes of each feature were calculated and averaged across the 10 percentile bins. These averaged effect sizes represent expected effect sizes based on discriminatory power of representative features from each percentile ranking. Power was then calculated from Cohen’s d effect sizes of the selected features to find the difference in means (post–pre measurement) between the Aronia and placebo group. These power values are therefore representative of the expected discriminatory power from different ranks of features. Using this method, the sample size of *n* = 14 had very high power (0.87 to 0.99) for the top 10 percentile of ranked features, which had a large average effect size (d = 0.93). Power was additionally calculated a posteriori from the linear regression model for fasting total cholesterol using the pwr.f2.test in the pwr package in R. The computed power was 97% at a type 1 probability of 0.05.

#### 2.18.2. Anthropometric, Blood Marker, and Inflammation Biomarker Analysis

Descriptive summary statistics were determined for participants’ physical and biological characteristics and general linear models were used to determine if the characteristics differed between the intervention groups at baseline. The impact of the intervention on the following categories of variables was assessed using general linear models: anthropometric measures (visceral adipose, fat mass percent, weight); fasting glycemic measures (HbA1c, GLU); fasting serum lipid measures (LDL, HDL, CHOL, TG); and fasting inflammation markers (TNF-α, GM-CSF, IFN-γ, IL-23, IL-10, IL-1β, IL-17, and IL-6). Normalization of cytokines was performed using the R package bestNormalize [50] prior to statistical analysis. All dependent variables were summarized as a delta value calculated as the change in value from baseline (post-intervention–pre-intervention). Predictor variables in linear models were common within each variable category as follows: anthropometrics–intervention group; blood lipid profile–intervention group, visceral adipose tissue; glycemic control –intervention group; inflammation–intervention group, visceral adipose tissue. DHQ III HEI scores were analyzed through general linear models to assess habitual diet and to identify any dietary differences between intervention groups at baseline.

Postprandial metabolic (GLU, TG) and inflammatory (TNF-α, GM-CSF, IFN-γ, IL-23, IL-10, IL-1β, IL-17 and IL-6) measurements collected at time points 0 (fasting) and 1, 2, 4 and 6 h postprandially were summarized as net area under the curve (AUC) using the auctime [51] R package. Each dependent variable was then summarized as a delta value representing the change in AUC (post-intervention—pre-intervention). Inflammation measures were normalized using the R package bestNormalize prior to statistical analysis. The impact of the intervention on each dependent variable was assessed using general linear models with identical predictor variables to those utilized in the corresponding fasting models.

#### 2.18.3. Fecal Microbial Statistical Analysis

A total of 738,952 raw reads were obtained across all samples. To aid unbiased diversity matrices due to sequencing depth, data were randomly subsampled in MOTHUR to the minimum number of sequences across samples (6001 reads/sample). Subsampling resulted in a total of 168,028 high-quality reads.

We performed ecological analyses and visualizations with RStudio (version 2023.12.0.369) running base R 4.3.2 unless specified differently. Alpha diversity (richness and Pielou’s evenness) was calculated at the OTU level using MOTHUR (version 1.35.1) (mothur.org) and vegan (version 2.6-4). Beta-diversity analyses were performed on subsampled data with filtering of OTUs less than 3 counts in at least 20% of the samples. Permutational multivariate analysis of distance matrices, or PERMANOVA, was performed with 999 permutations using adonis2 in vegan (version 2.6-4). Fixed effects included an interaction between time and juice condition, underlying main effects in the interaction, and subject. We examined community composition at each taxonomic level (phylum–genus) in addition to taxa selected by hierarchical feature engineering using taxaHFE without the permuted random forest [52]. Beta diversity was visualized using non-metric dimensional scaling from phyloseq [53].

A differential abundance analysis was also performed to identify possible changes in genera from four weeks of juice consumption. We utilized MaAsLin2 [54] with the subject as the random effect and an interaction between time and juice condition as our fixed effect. Default MaAsLin2 parameters were utilized. We then extracted the interaction output and re-calculated the q-values for multiple testing adjustment with the Benjamini–Hochberg method.

#### 2.18.4. Metabolomics Statistical Analysis

Following LCMS analysis, serum data were converted to mzML format using MSConvert v3.0 (proteowizard.sourceforge.io). The data were then mined using mzMine v3 and MS-DIAL [55] (mzmine.github.io) with an intensity cut-off of 1000, a retention time window of 0.25 min, and an error of 15 ppm for isolation of unique features. Solvent and processing blanks were used to remove background noise and retain biologically relevant metabolites, where features were retained if they had an area greater than five times the processing and solvent blank. After data processing, over 950 features were retained within the serum samples. Fecal data were processed using Progenesis with a 5 ppm error identification search parameter. Almost 14,000 features were found in the fecal samples after removal of background noise in a similar manner to the serum dataset. Statistical analysis was completed using MetaboAnalyst v5.0 [56] (metaboanalyst.ca). Statistical analysis preprocessing steps included normalization via log transformation and filtering based on interquartile range. Statistical analyses including *t*-tests and ANOVAs were used to determine significant features, which were screened using the false discovery rate (FDR) in the untargeted analysis to correct for multiple testing. Identified significant features were examined as delta values (post-intervention relative concentration–pre-intervention relative concentration).

## 3. Results

### 3.1. General Characteristics of Participants

A total of 25 adults were screened for study participation, of which 10 were excluded for not meeting inclusion criteria, lost to follow-up, or withdrew for non-intervention related reasons. Fifteen individuals were randomized to an intervention group, with one participant lost for phlebotomy difficulties (Appendix A). A total of 14 adults (mean ± SD); age (32.87 ± 6.98, years); body mass index (BMI) (25.75 ± 4.66 kg/m^2^) completed the study. One participant who completed the study was excluded from lipid panel (GLU, TG, CHOL, HDL, LDL) and inflammation blood analyses due to acute alcohol consumption prior to the last blood collection visit but was included in the metabolomic and microbial composition analysis. 

The participant cohort generally exhibited good metabolic health (Table 2). On average, participants in both intervention groups were mildly overweight as classified by BMI and had normal blood pressure as defined by the American Heart Association guidelines [57]. Fasting blood GLU and HbA1c were normal on average. Fasting total CHOL, LDL cholesterol, and TG were normal and HDL was intermediate in both intervention groups as compared to the standard reference range. Anthropometric and blood measures were similar between groups at baseline.

### 3.2. Anthropometric Measures

No differences were observed between ARO and PLA in the change in fat mass percentage (F = 1.02, *p* = 0.34), body weight (F = 0.09, *p* = 0.77), or visceral adipose tissue (F = 1.23, *p* = 0.29) over the 30-day intervention.

### 3.3. Habitual Diet Analysis

Participants in ARO and PLA had similar habitual diets at baseline as determined by comparisons of HEI total and component scores from the self-reported DHQ III questionnaire (Appendix A). HEI component and total HEI scores were not different by intervention group with the exception of saturated fat intake, which was lower in the ARO group (t = −2.54, *p* = 0.03) (Appendix A). Importantly, HEI component scores for categories associated with high polyphenol content including total vegetables (F = 0.03, *p* = 0.87), greens and beans (F = 0.25, *p* = 0.62), total fruit (F = 1.88, *p* = 0.20), and whole fruit (F = 0.46, *p* = 0.51) were not different between intervention groups. Along with fruits and vegetables, caffeinated beverages such as coffee and tea are major dietary sources of polyphenol intake [12]. Participants were not instructed to avoid these beverages during the intervention, but caffeine intake (mg/day) was not different between groups (F = 0.01, *p* = 0.95) at baseline. However, there are other polyphenols in coffee and tea in addition to caffeine that are not estimated by the DHQ III questionnaire. Although not measured directly, the congruence in HEI component scores and caffeine intake between intervention groups suggests a comparable habitual intake of polyphenols.

### 3.4. Fasting and Postprandial Lipid and Glycemic Measures

Fasting blood lipid panels were completed pre- and post-intervention to determine if markers were modulated by Aronia intervention. We found that Aronia consumption was associated with stable total CHOL (β = −0.50, *p* = 0.03) levels over the 30-day intervention period compared to the placebo group (Figure 1). ARO had a small increase of 0.08 mmol/L, representative of maintained baseline levels, whereas PLA averaged an increase of 0.40 mmol/L. No difference was detected between intervention groups in fasting GLU (F = 3.82, *p* = 0.08), TG (F = 0.44, *p* = 0.52), HDL (F = 2.39, *p* = 0.15), or LDL (F = 3.13, *p* = 0.11).

To determine the impact of Aronia consumption on postprandial metabolism, we examined triglyceride and glucose responses to a high-fat meal challenge. There was no effect of Aronia (F = 1.19, *p* = 0.30) on the change in TG AUC (dTG_AUC_), with an average dTG_AUC_ of 1.27 mmol/L in ARO and 0.10 mmol/L in PLA (Appendix A). There was substantial variation between individuals in the dTG_AUC_ in both intervention groups, with responses ranging from a 0.91 mmol/L decrease to a 4.97 mmol/L increase from pre- to post-intervention. A summary of the postprandial triglyceride response by intervention group is provided in Appendix A.

Aronia consumption was associated with lowered postprandial blood glucose responses (β = −3.03, *p* < 0.01), with an average decrease of 1.94 mmol/L compared to a 1.09 mmol/L increase in the PLA group (Figure 2). All participants maintained blood glucose levels under 7.8 mmol/L at the 2-hour postprandial time point, indicating a normal blood glucose response to a mixed meal. A summary of the postprandial blood glucose response before and after the intervention is provided in Appendix A.

### 3.5. Fasting and Postprandial Inflammation

As high-fat meals can induce an inflammatory response [58], we examined the impact of 30-days of Aronia on postprandial inflammation in addition to fasting levels. We did not detect an effect of intervention group on changes in fasting or postprandial inflammation markers from pre- to post-intervention (Appendix A).

### 3.6. Fecal Microbial Alpha and Beta Diversity

We examined the impact of four weeks of juice consumption on fecal microbial richness and evenness (alpha diversity) and community composition (beta diversity). We found that the fecal microbial community of individuals who consumed Aronia juice had an average decrease in microbial richness (β = −18.2, *p* = 0.04, Figure 3A), accounting for BMI (β = −0.31, *p* = 0.73). We did not detect a difference in microbial evenness as measured by Pielou’s evenness between the Aronia and placebo groups (*p* = 0.74, Figure 3B). We also did not observe a difference in the microbial community composition at any taxonomic level or with microbial features selected through hierarchical feature engineering after the intervention (Appendix A). Unconstrained ordination at the genus level is provided in Figure 3C.

### 3.7. Differential Microbial Taxa

With the decrease in microbial richness, we also performed a differential abundance analysis using MaAsLin to examine what microbial genera may have changed in abundance after four weeks. Without multiple testing correction, we detected a decrease in the abundance of *Holdemania* and *Barnesiella* with Aronia consumption and increased abundance of three unidentified genera within *Oxalobacteraceae*, *Prevotellaceae*, and *Pasturellaceae* (Appendix A). These five associations were not maintained after multiple testing correction.

### 3.8. Serum Metabolomics

Following LCMS data acquisition and processing (blank and noise removal, peak alignment) the serum dataset consisted of ~950 features. An initial analysis of pre- and post-intervention samples from both the placebo and Aronia groups by PCA 2-D plot, demonstrated the similarity of the metabolomic profiles and the overall data quality and reproducibility (Appendix A).

Annotation using authentic standards, fragmentation library matching, and in silico annotation resulted in 47 unique identifications (Appendix A). Delta values for each identified metabolite were determined by subtracting the measured pre-intervention relative concentration from the post-intervention relative concentration. A *t*-test was then performed for each metabolite and two metabolites were found to have significant *p*-values (<0.05) after FDR correction (Figure 4). This analysis indicated that fasting levels of both asparagine and tyrosine were significantly different, with both showing an increase in concentration in the ARO group after the intervention.

An untargeted analysis was also completed for the fasting serum metabolomics dataset. This analysis found three features to be significantly different (*p* < 0.05) between the first and the final blood collection (Figure 5A). These features included *m*/*z* 120.002, 164.027, and 286.200. Annotations for these features were made using mass values and tentative annotations were made for 164.027 as 2-chloroquinoline and 286.200 as 6-octenoylcarnitine, while no reliable annotation for the other features could be made. 6-octenoylcarnitine was downregulated in ARO while the other two features were elevated with Aronia consumption (Figure 5B).

### 3.9. Fecal Metabolomics

The fecal LCMS analysis produced ~14,000 features after removing background noise. A global analysis of the fecal metabolomic profiles from both the control and experimental groups indicated similar metabolomic profiles, with no separation or grouping of participants (Appendix A). This again showed the reproducibility of the data and consistent LCMS results.

Authentic bile acid reference standards were used in conjunction with the fecal sample analysis to identify 15 bile acids (Appendix A). Statistical analysis was performed, and one bile acid changed significantly (*p* < 0.10) between the first fecal collection and the fecal collection four weeks later. The bile acid, chenodeoxycholic acid (CDCA), was upregulated in fecal samples from individuals receiving Aronia after four weeks (Figure 6).

An untargeted analysis of the acquired fecal data was also performed. Through fragmentation library matching and isotopic scoring, 42 level 2 annotations were made (Appendix A). Of these, four were found to be significantly different using a paired *t*-test (Figure 7A–D). All features found were lipids and were upregulated after four weeks of Aronia consumption.

## 4. Discussion

High-polyphenolic-content Aronia fruits have great potential as a functional food with established anti-inflammatory, hypolipidemic, and hypoglycemic biologic activities [59]. The beneficial impact of polyphenolics on metabolism are likely mitigated by fecal microbial metabolism of polyphenols, necessitating simultaneous assessment of systemic and gut metabolism to elucidate mechanistic underpinnings of health outcomes. We conducted a 30-day clinical trial to characterize the effects of Aronia intervention in human participants by analyzing fasting and postprandial inflammation, lipid, and glucose metabolism biomarkers as well as fecal microbial composition and fecal and serum metabolites. The key findings of this study include decreased postprandial glucose, attenuated fasting cholesterol, a reduction in microbial richness, and alterations to both gut and host metabolomes with Aronia juice consumption. Metabolomic changes are consistent with lowering of inflammation; however, resting and postprandial inflammatory cytokine levels were not changed. Further, withdrawal of many polyphenol sources in the background diet of the control group appears to have negatively impacted metabolism.

We found that 30-days of Aronia juice consumption was associated with lower fasting total cholesterol levels compared to a placebo-matched control. Improvements to blood lipid levels with polyphenolic intake are well demonstrated in the literature [11]. Several potential mechanisms for their cholesterol-lowering effect have been identified in murine models including inhibition of intestinal lipid absorption [60] and regulation of genes related to glycerophospholipid metabolism [61]. Few controlled trials have focused on Aronia-based interventions; however, two meta-analyses of randomized controlled trials investigating the impact of Aronia on cholesterol levels identified improved measures of total cholesterol [62,63] as well as LDL and HDL cholesterol [63]. It is possible that the significant difference in CHOL levels between the Aronia and placebo groups is reflective of the low-polyphenolic diet followed during the study, with the lack of polyphenolic intake in the placebo group contributing to increased CHOL levels. No other changes to blood lipid measures including HDL, LDL, and TG were identified in our study, which is consistent with results from a randomized crossover trial examining Aronia consumption in a healthy-participant cohort [14]. We are not aware of any other studies examining changes to the postprandial triglyceride response after long-term Aronia consumption to provide context for our study measure indicating no change. However, this finding may be reasonably expected given the metabolically healthy status of our participants and the high inter-individual variability in their responses both before and after the intervention.

Aronia juice consumption lowered postprandial glucose responses to the high-fat meal challenge. In a similar study, 100 mL of Aronia juice was found to lower postprandial blood glucose in healthy subjects [64]. This decrease was attributed to inhibited enzymatic activity of dipeptidyl peptidase IV, α-glucosidase, and angiotensin-converting enzyme by Aronia juice. Inhibition of these enzymes reduces blood glucose levels through different mechanisms including incretin regulation and epithelial glucose transport [65,66,67]. However, blood glucose levels were measured in response to acute Aronia juice ingestion, which differs from our study as no juice was consumed on the day of high-fat meal testing. Therefore, it is possible that a separate mechanism is responsible for the lowered blood glucose responses in our study. Several factors impact postprandial glycemic responses include baseline glycemic status and insulin secretion. We did not measure fasting or postprandial insulin responses, which prevents us from speculating on the relationship to insulin secretion in our study. However, the baseline glycemic status of our participants was optimal, as measured by fasting blood glucose and HbA1c, and no changes to these metrics were identified in the Aronia group. As identified with total cholesterol levels, small increases in postprandial glucose were identified in the placebo group. Again, this result may be due to the removal of polyphenolics from the background diet. More studies examining postprandial glucose responses to long-term Aronia consumption are needed to better define the beneficial impacts of Aronia on postprandial glycemic control.

Contrary to our hypothesis, fasting inflammation levels were not decreased after 30 days of Aronia juice consumption. Randomized controlled trials have reported lowered inflammation [14,68] with Aronia supplementation, though some found no change [18]. These studies were conducted in metabolically at-risk populations and had intervention periods lasting 6–12 weeks. Therefore, it is possible that fasting inflammation markers were unchanged in our study due to a much shorter intervention period and the use of metabolically healthy participants as the study cohort. Our study was designed to detect systemic anti-inflammatory changes induced by Aronia, rather than acute influences. However, as postprandial samples were collected for triglyceride and glucose measurements and high-fat meal challenges are an established inflammatory stimulus, we also completed a postprandial inflammatory analysis. No improvements to postprandial inflammation markers were observed with Aronia consumption. This is likely because maximal serum concentrations of polyphenols were not aligned with the timing of the high-fat meal, the inflammatory stimulus used in this study, for greatest anti-inflammatory benefit. Hydroxycinnamic acids and anthocyanins from Aronia are partially absorbed in the small intestine and reach their maximal serum concentrations within three hours of consumption [69,70]. However, 90–95% of polyphenols pass to the large intestine and are subject to digestive metabolism by gastrointestinal microbes. These microbial derivates of the parent polyphenols have a delayed entry into the circulation. Consumption of isotope-labeled cyanidin-3-glucoside in healthy adults revealed that derivates ferulic and hippuric acid, respectively, take an average of 8.2 and 15.7 h to reach maximal plasma concentrations [71]. Furthermore, some anthocyanin metabolites have a half-life of up to 96 h [71]. Therefore, it is possible that some bioactive derivatives were in circulation during challenge meal testing. However, plasma concentrations were likely a small fraction of peak concentrations after acute ingestion. The authors hypothesize that the timing of juice consumption and challenge meal testing may also have contributed to the lack of change to postprandial triglyceride responses with Aronia consumption.

Our metabolomic exploration to identify metabolic underpinnings driving the health impacts of Aronia uncovered changes that may reflect fundamental metabolic shifts. Suspect screening for common metabolites in the serum dataset isolated several metabolites and the subsequent analysis indicated changes in serum amino acids with Aronia consumption; specifically, upregulation of fasting concentrations of asparagine and tyrosine. These amino acids, among other amino acids, are central to a myriad of metabolic pathways including energy metabolism and various signaling pathways. While challenging to elucidate the exact cause and function of these increases, this phenomenon has been previously reported with phenolic supplementation in vitro. Gerdemann et al. found that polyphenolic supplementation specifically upregulated asparagine and tyrosine, among other amino acids [72]. This consistent result indicates the role that Aronia could have on central carbon metabolism and other vital metabolic pathways.

While fasting and postprandial inflammatory cytokines were not altered by Aronia consumption, we identified potentially anti-inflammatory changes in the metabolome. Lower serum concentrations of 6-octenoylcarnitine were identified in participants consuming Aronia juice. 6-octenoylcarnitine is a medium-chain acylcarnitine formed by the oxidation of longer-chain acylcarnitines [73]. Medium-chain acylcarnitines have been shown to activate pro-inflammatory pathways [74]. 6-octenoylcarnitine has been associated with inflammatory pathways [75] and is elevated in populations with diseases such as diabetes and multiple sclerosis [76,77].

Having established systemic metabolic impacts of Aronia in circulating biomarkers and metabolites, it was anticipated that changes in the composition of the gut microbiome or microbial metabolism were likely. We found that Aronia consumption reduced microbial richness, an aspect of alpha diversity, without affecting the beta diversity of the gut microbiome. It is believed that a high alpha diversity in the context of the gut provides functional redundancy to help ensure continued specialized metabolic functions [78]. A reduction in richness could be indicative of an antimicrobial effect from high-polarity phenolic derivatives, though this effect seems less likely as we did not observe differentially abundant bacteria after multiple testing correction. It is also possible that refraining from polyphenol-rich foods during the intervention contributed to reduced richness. Since polyphenols co-occur with other nutrients such as fiber, omitting polyphenol-rich foods may have limited the habitual dietary substrate pool for growth of the established microbial community [79]. It is unlikely that the introduction of Aronia juice as an addition to a good habitual diet in healthy adults would shift the fecal microbial community. However, we have previously seen that in times of large microbial disturbances, such as with a high-fat diet, that Aronia juice may prevent further declines in alpha diversity [22]. Taken together, the lack of community-wide changes and specific microbial signals suggests that Aronia polyphenols are metabolized by a range of microbial taxa in the gut microbiome of healthy adults. Further study is needed to assess the metabolic capacity of the human gut microbiome to metabolize anthocyanins and how resulting phenolic derivatives may influence postprandial outcomes.

Bile acids in the gut have the potential to influence glucose and lipid metabolism as well as inflammation, and thus, were of specific interest in this investigation [80]. Our analysis indicated that CDCA, a primary bile acid, was upregulated after 4 weeks in study participants receiving Aronia. CDCA facilitates lipid and glucose regulation through several actions including activating the farnesoid X receptor [81]. Previous work has demonstrated that CDCA reduces pro-inflammatory cytokines and can improve lipid profiles while reducing oxidative stress [82,83]. CDCA levels are also associated with gut microbiome changes, including increases in *Akkermansia*, *Bifidobacterium*, *Lactobacillus*, and Desulfovibrio populations [84]. From an untargeted analysis of fecal metabolites, we detected four lipids with different delta relative concentrations between groups. This could be due to the effects of Aronia, possibly even due to the significant changes induced in CDCA or via differential microbial metabolism.

While our study demonstrated several important impacts of Aronia juice consumption in humans, we also acknowledge study limitations. Our experiment was set up as a feasibility study to explore the impacts of Aronia on measures of metabolic health in humans. The size of our participant cohort was a result of this study design and we acknowledge that a greater sample size is needed to confirm and further characterize metabolic responses to Aronia. The study cohort was also characterized as metabolically healthy at baseline, which likely indicates a smaller effect size for study measures in our participants compared to individuals with metabolic risk factors. We did not assess the absorption or metabolism of Aronia, as pharmacokinetic analysis was not a focus of this study. Future research in this area could offer valuable insights into the link between Aronia-derived polyphenolics and health outcomes. In addition, although participants were instructed to maintain their normal physical activity patterns while enrolled in the study, alterations to exercise activities prior to bloodwork collection visits could impact study measures. Non-compliance with the avoidance of high-polyphenol-content foods during the intervention could also impact study results. While participants were asked to record instances when they consumed a food item from the high-polyphenolic food item list provided, they were not excluded from the study for doing so. Additionally, while caffeinated beverages such as tea and coffee are major dietary sources of polyphenols [12], participants were not instructed to limit intake of these items during the intervention period. This decision was purposefully made to improve study feasibility and participant recruitment. For these reasons, it is possible that polyphenolic intake from diet during the intervention was different between groups and impacted the study results. In addition, acute dietary intake prior to sample collection was not recorded, and therefore, cannot be controlled for in this study.

Our study uniquely characterizes Aronia-induced metabolic impacts in a healthy human cohort. Although the benefits of polyphenol intake are known to be mitigated, at least in part, by the gut microbiome, studies examining changes in microbial taxonomy and metabolism in response to Aronia consumption in humans are limited [19]. In addition to fasting markers of metabolic health, we measured postprandial lipid, glucose, and inflammation responses to a high-fat meal challenge. While the importance of postprandial metabolism in relation to metabolic health and disease risk is well established [30,31,32,33,34], these measures are not consistently analyzed in non-acute studies. Inclusion of these measures in our study more comprehensively characterizes the impact of our Aronia dose on metabolic health. Other study strengths include the use of a whole-food-based intervention in free-living adults, which best characterizes dietary changes likely to be made by individuals seeking to improve metabolic health.

## 5. Conclusions

Thirty days of Aronia juice consumption prevented increases in total cholesterol and improved postprandial glucose responses. Metabolomic analyses suggested potential shifts in systemic metabolism and decreases in inflammatory metabolites, but no clear link to the measured impacts. A decrease in microbial alpha diversity, as measured by microbial richness, was observed, which was likely a side effect of dietary restrictions during the intervention to isolate the impact of Aronia-derived polyphenols. The beneficial changes to host metabolism identified in this study indicate that Aronia juice consumption may be an effective dietary strategy to improve metabolic health. Further clinical trials investigating Aronia-mediated health impacts are warranted and may consider a larger dose or longer intervention period to further identify potential metabolic benefits of Aronia. This information further informs study design for future clinical trials and provides novel insight needed for the development of polyphenolic-based dietary strategies to improve health.

## Figures and Tables

**Figure 1 foods-13-02768-f001:**
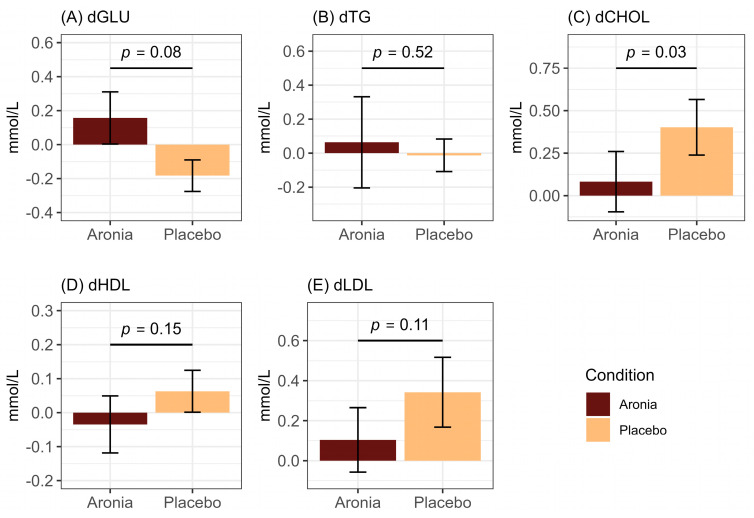
Summary plots of delta (d) values calculated as (post-intervention−pre-intervention) in (**A**) fasting glucose (dGLU), (**B**) fasting triglyceride (dTG), (**C**) fasting total cholesterol (dCHOL), (**D**) fasting HDL cholesterol (dHDL), and (**E**) fasting LDL cholesterol (dLDL). Differences between intervention groups determined by ANOVA.

**Figure 2 foods-13-02768-f002:**
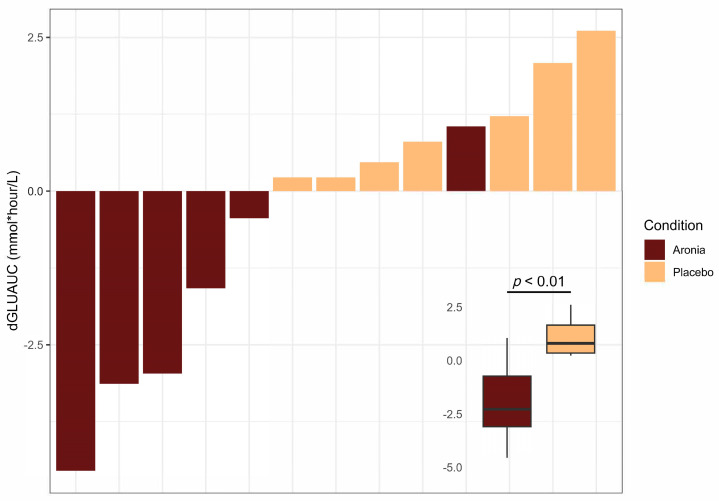
Summary plot of individual changes (post-intervention−pre-intervention) in glucose area under the curve (dGLU_AUC_). dGLU_AUC_ values represent the change in the sum of values from fasting and hourly time points for 6 h post-high-fat meal ingestion. Each bar is representative of the value for one participant (*n* = 13). Difference in dGLU_AUC_ between meal groups determined with ANOVA and displayed in inset.

**Figure 3 foods-13-02768-f003:**
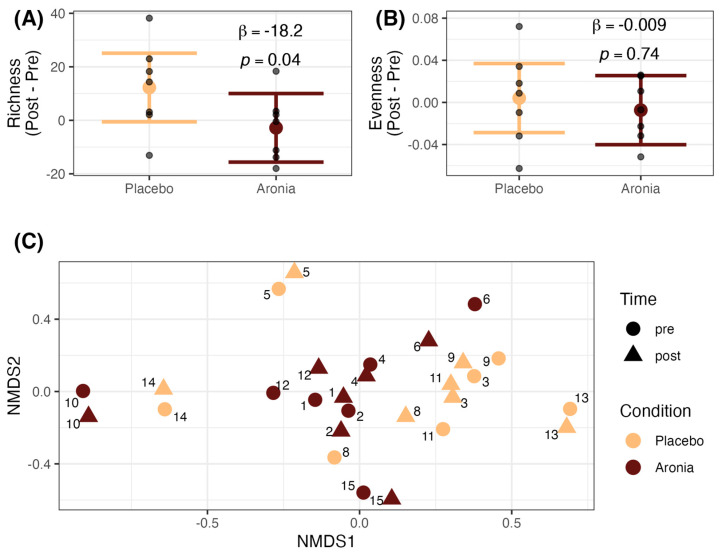
Alpha and beta diversity of the gut microbiome. (**A**) Microbial richness, (**B**) microbial evenness, and (**C**) microbial community composition. Richness and evenness beta coefficients were derived from non-parametric rank-based estimation methods. Bars represent 95% confidence interval. Beta diversity at the genus level was visualized using non-metric dimensional scaling (NMDS). Each point represents a single sample, with numbers indicating the specific subject.

**Figure 4 foods-13-02768-f004:**
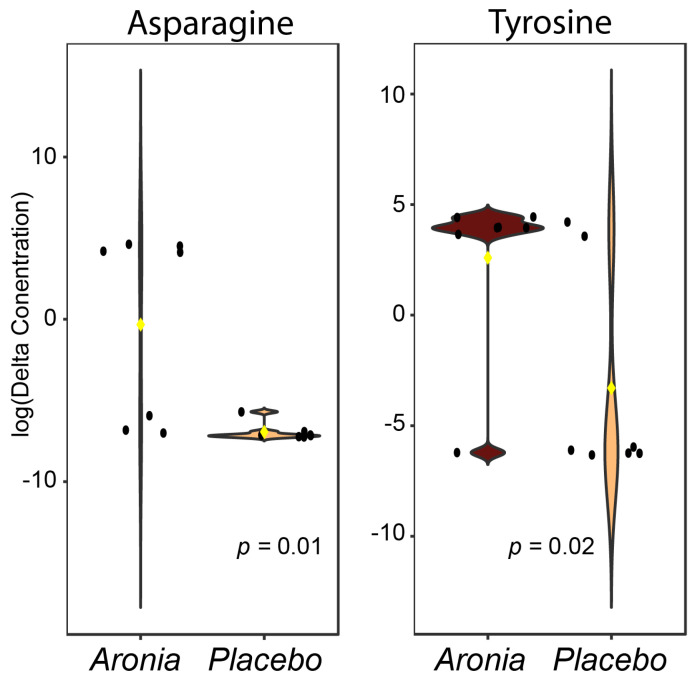
Suspect screening statistical analysis results in serum. Violin plots depict relative concentration changes (post−pre intervention) of fasting asparagine and tyrosine levels. Black points represent participant values and yellow rhombus indicates group median. Both amino acids were upregulated in the Aronia group after 4 weeks. *p*-values are indicated in each panel for the relationship using a *t*-test.

**Figure 5 foods-13-02768-f005:**
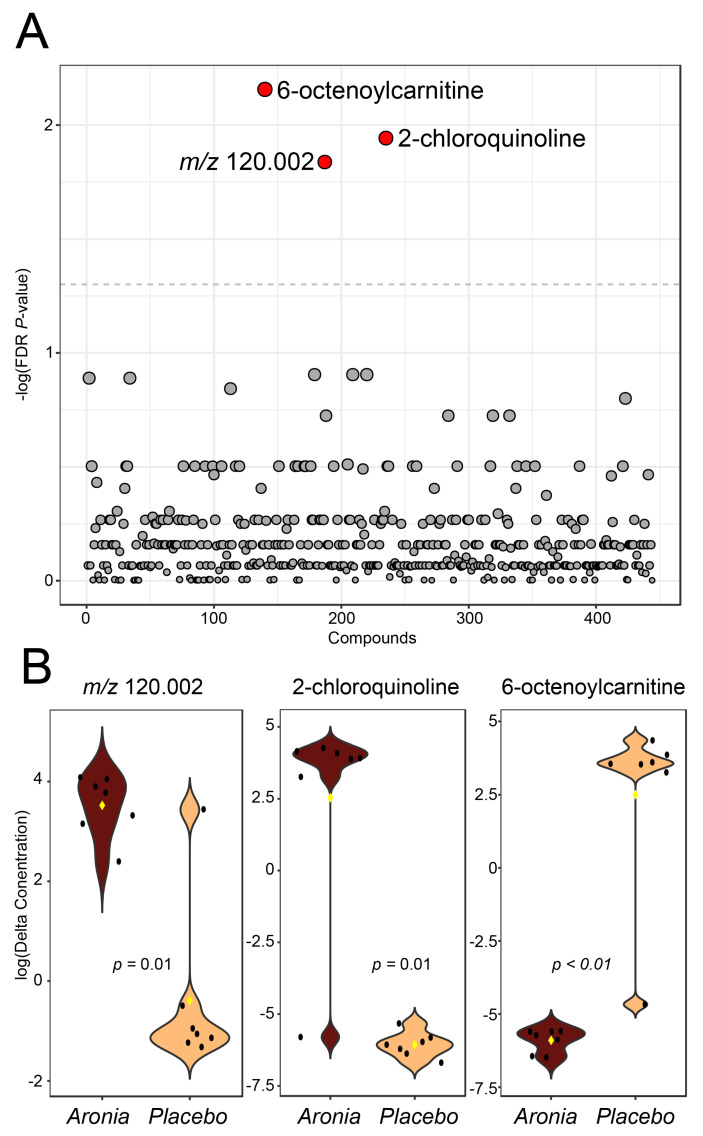
Untargeted analysis of the fasting serum metabolome. (**A**) *t*-test results comparing relative concentration change (post−pre intervention) of serum metabolome features between experimental groups. Features with *p* < 0.05 after FDR correction are indicated with red dots and features above this cut-off are indicated with gray dots. (**B**) Violin plots of the three significant features, with specific *t*-test *p*-values shown. Black points represent participant values and yellow rhombus indicates group median.

**Figure 6 foods-13-02768-f006:**
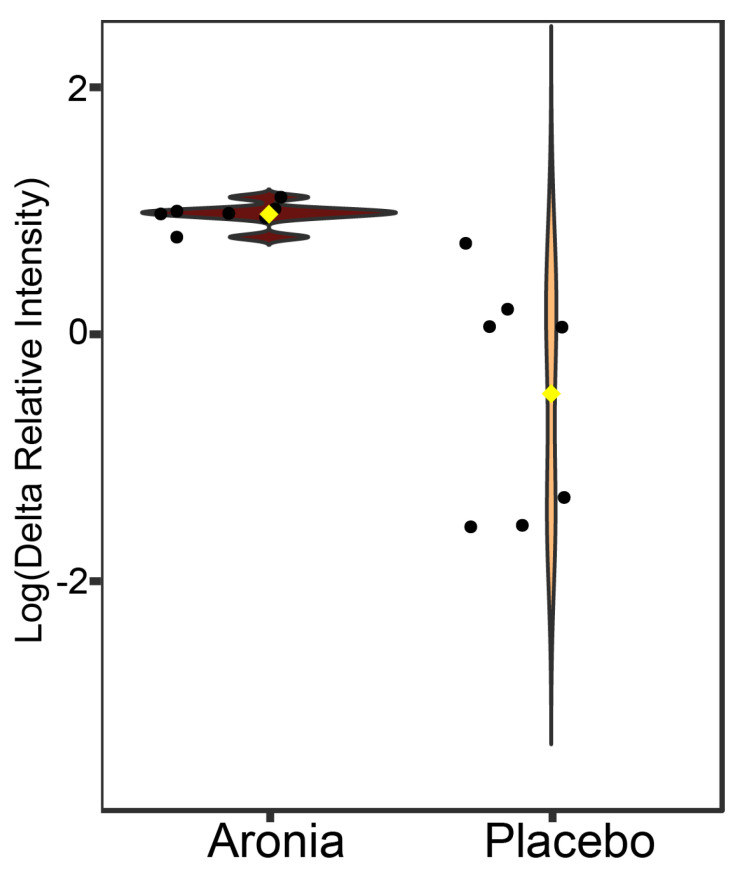
Violin plot of the relative concentration change (post–pre intervention) of fecal chenodeoxycholic acid (CDCA). Black points represent participant values and yellow rhombus indicates group median. CDCA levels were upregulated in the Aronia group after 4 weeks. A *p*-value of 0.07 was calculated using a one-way ANOVA test.

**Figure 7 foods-13-02768-f007:**
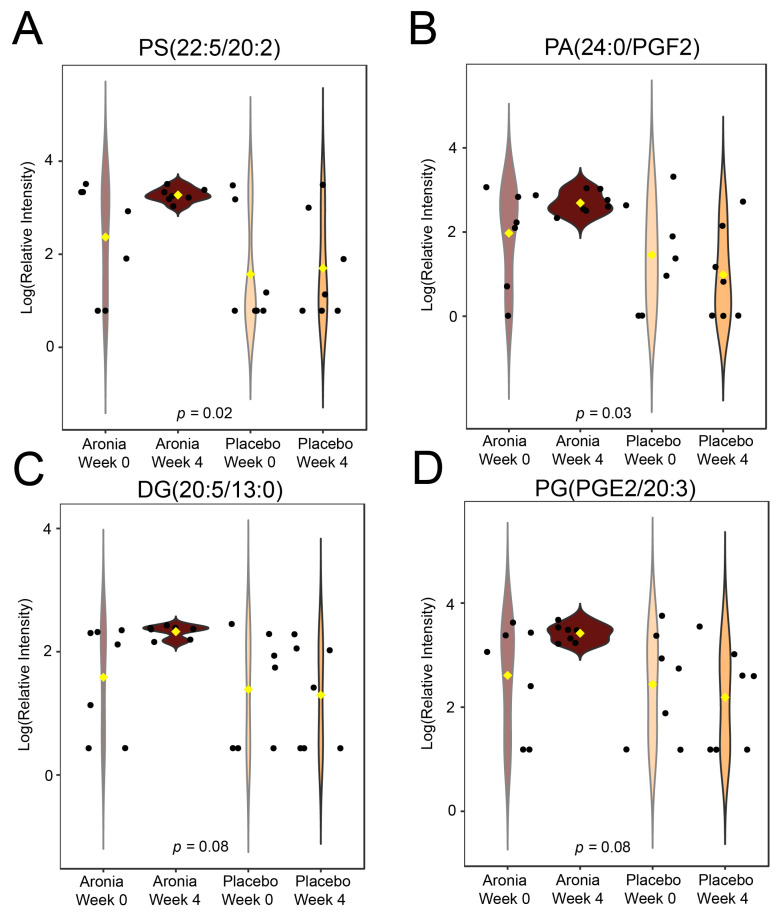
Untargeted analysis of fecal metabolome. Violin plots (**A**–**D**) depict the relative concentrations of significantly different features before and after the intervention in both experimental groups. Black points represent participant values and yellow rhombus indicates group median. All annotated features were found to be lipids: phosphatidylserine (PS), phosphatidic acid (PA), diglyceride (DG), and phosphatidylglycerol (PG).

**Table 1 foods-13-02768-t001:** Carbohydrate and polyphenol content of Aronia juice.

Compound	Concentration
Carbohydrate	(g/100 mL)
Fructose	7.79
D-glucose	7.45
Sorbitol	12.85
Polyphenol	(mg/100 mL)
Phenolic Acids	
Chlorogenic acid	304.66
Neochlorogenic acid	318.44
Anthocyanins	
Cyanidin 3-xyloside	1.94
Cyanidin 3-glucoside	3.20
Cyanidin 3-galactoside	28.54
Cyanidin 3-arabinoside	51.34
Flavonols	
Quercitin 3-galactoside	0.89
Quercitin 3-rutinoside	2.04
Quercitin 3-glucoside	5.91

**Table 2 foods-13-02768-t002:** Baseline participant characteristics by assigned dietary intervention (*n* = 14). Blood profile represents fasting baseline value.

	Placebo (*n* = 7)	Aronia (*n* = 7)	*p*-Value
Sex (male/female)	3/4	3/4	0.92
Age (years)	32.4 ± 7.0	35.0 ± 7.8	0.55
BMI (kg/m^2^)	25.5 ± 4.0	26.4 ± 6.2	0.77
Fat mass (%)	25.5 ± 13.4	29.5 ± 10.7	0.57
Visceral adipose (L)	0.84 ± 0.93	1.77 ± 2.1	0.34
Blood pressure (mmHg)			
Systolic	111.00 ± 20.78	108.88 ± 15.71	0.84
Diastolic	67.83 ± 18.23	67.00 ± 10.61	0.92
HbA1c (%)	4.87 ± 0.34	5.05 ± 0.35	0.37
Fasting glucose (mmol/L) ^1^	5.33 ± 0.31	5.11 ± 0.46	0.34
Triglycerides (mmol/L) ^1^	1.03 ± 0.41	1.41 ± 1.46	0.52
Total cholesterol (mmol/L) ^1^	4.29 ± 0.62	4.46 ± 0.88	0.70
LDL cholesterol (mmol/L) ^1^	2.81 ± 0.72	2.69 ± 0.42	0.71
HDL cholesterol (mmol/L) ^1^	1.40 ± 0.38	1.36 ± 0.55	0.90

Data represent mean ± standard deviation. Characteristic marked with ^1^ indicates *p*-value calculated with (*n* = 13) due to exclusion of one participant for not meeting fasting guidelines. *p*-value for sex determined by two-sample test for given proportions; all other *p*-values determined by a two-sample *t*-test. Abbreviations: body mass index, BMI; glycated hemoglobin, HbA1c; low-density lipoprotein, LDL; high-density lipoprotein, HDL.

## Data Availability

The metabolomic and 16S gut microbial taxonomy datasets presented in this study can be found in online repositories. Metabolomic data including raw files may be found at https://www.metabolomicsworkbench.org. 16S microbial taxonomy data may be found at http://www.ncbi.nlm.nih.gov/bioproject/1136926., PRJNA1136926. Other data presented in this study are available on request from the corresponding author due to participant confidentiality.

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
