# Peer review of "Polyphenol-Rich Aronia melanocarpa Fruit Beneficially Impact Cholesterol, Glucose, and Serum and Gut Metabolites: A Randomized Clinical Trial"

_foods, 2024, doi:10.3390/foods13172768_

Round 1

Reviewer 1 Report

Comments and Suggestions for Authors

The clinical study describes the health benefits of Aronia fruit using different assessments. However, there are several areas that need improvement. The trial doesn’t even fit the criteria of a pilot study according to NIH guidelines. The polyphenols from Aronia fruit are highlighted at several places in the manuscript but no data is provided on the absorption/metabolism of these compounds after ingestion for 30 days. Including that data will strengthen the manuscript. The manuscript can be accepted for publication after authors address the comments provided below.

1.      The title should reflect it as a feasibility study.

2.      Add a sentence about statistical analysis in the abstract

3.      Line 65: Please cite original research papers not review articles.

4.      Lines 87-88: There are already couple of clinical trials on Aronia berries. Some with longer duration than your study. Modify the sentence

5.      Lines 109-110: previous smokers? alcohol consumption, tea, coffee drinkers? please provide more details on the inclusion/exclusion criteria

6.      Lines 150-152: why it is called supplement intervention? Aronia juice is not a supplement. Also provide details if the juice was prepared in house or purchased from a company.

7.      Table 1: Provide placebo details in the same table for comparison.

8.      Line 343: why blood bio-markers and inflammatory markers were not taken into account for power analysis? Authors should clarify their objectives in the introduction.

9.      Lines 384-385: Rewrite the sentence.

10.   Lines 444-445: These numbers are not normal according to the latest AHA guidelines. Update with a new reference.

Reviewer 2 Report

Comments and Suggestions for Authors

The article has poor innovation, and there are still a lot of problems in the manuscript that need to be solved.

1.  L28 PLA and L29 ARO need to be full expanded at first mention. Please check the full text.

2.   L121 The components of a high-fat meal should also be provided.

3.  While it is good to use people as test subjects, the randomness and uncontrollability of the data also increases. The sample size of subjects is too small and the age span of subjects is too large. Greater sample sizes are needed to confirm and further characterize metabolic responses to supplementation.

4. The data in the manuscript is lack of innovation and depth. It is suggested to re-analyze the data and present the results.

5.  Please, check the format of tables and figures according to journal requirements. And, the abbreviations used in the tables and figures should be explained in the footnote of tables and/or figure captions to make them standalone.

6. Section 4 - The discussion on results has not been extensively performed, which is crucial to highlight any possible promising applications, especially correlating the experimental data and results obtained by different analytical methods.

7.   Please, change p valueor pto P valueor P. And check the format of formulas according to journal requirements.

8.  Revise carefully reference style. For example, volume (number included or not), Journal name (italic or not), etc.

Comments on the Quality of English Language

Minor editing of English language required.

Reviewer 3 Report

Comments and Suggestions for Authors

This research examined the effect of a polyphenol rich fruit (Aronia) on inflammation, cardiometabolic risks, and gut microbiome in apparently healthy individuals.  The manuscript is well-written and provides an in-depth analysis of the outcomes.  Well done!

Minor comments

1.  Research design paragraph - briefly mention what the placebo is.

2.  Research design paragraph - what are a few examples of high polyphenol foods

3.  Research design paragraph - When were samples and measurements with regards to consumption of the intervention beverage?  

4.  Research design paragraph - maybe make it more clear about this being a study to assess chronic, resting impact and not the acute effect of the juice

5.  Anthropometrics paragraph - line 134-5, BIA is not used to assess BMI, but this was likely done using stadiometer and scale

6.  Line 381 - change word "makers" to 'markers"

7.  Line 469 - consider statement that while caffeine is a polyphenol, it is important to recognize there are other polyphenols in coffee and tea besides caffeine.  

8.  Was blood taken prior to the meal challenge, i.e. a baseline measure?  How was this in relation to the most recent dose of the juice? 

Round 2

Reviewer 2 Report

Comments and Suggestions for Authors

1.The data in Figure 3 require ANOVA (e.g. adding error lines). And the horizontal coordinate needs to be named.

2. Figure 5 and 7 should be given as clear picture.

3.Confirm the format of “p value” to meet recommendations made in MDPI Layout Style Guide (the style guidelines set forth by Foods), and check the full text.
